# LLM-IQA: Standard-guided MLLM for Mix-grained Image Quality Assessment

## Abstract

Image quality assessment (IQA) serves as the golden standard for all models' performance in nearly all computer vision fields. However, it still suffers from poor out-of-distribution generalization ability and expensive training costs. To address these problems, we propose **LLM-IQA**, a standard-guided zero-shot mix-grained IQA method, which is training-free and utilizes the exceptional prior knowledge of multimodal large language models (MLLMs). To obtain accurate IQA scores, namely scores consistent with humans, we design an MLLM-based inference pipeline that imitates human experts. In detail, LLM-IQA applies two techniques. **First**, LLM-IQA objectively scores with specific standards that utilize MLLM's behavior pattern and minimize the influence of subjective factors. **Second**, LLM-IQA comprehensively takes local semantic objects and the whole image as input and aggregates their scores, leveraging local and global information. Our proposed LLM-IQA achieves state-of-the-art (SOTA) performance compared with training-free methods, and competitive performance compared with training-based methods in cross-dataset scenarios. Our code will be released soon.

## 1 Introduction

Image quality assessment (IQA) aims to provide accurate quality scores that align with human mean opinion scores (MOS). With the booming of digital technology, the explosion of visual content calls for advanced IQA methods in all fields including communication (Zhou & Wang, 2022), entertainment (Wu et al., 2024e), professional use (Chow & Paramesran, 2016; Fang et al., 2020), and recently popular AI-generated content (Kirstain et al., 2023; Li et al., 2023). Over time, significant contributions have been made in this domain, evolving from traditional hand-crafted feature-based approaches (Wang et al., 2004; Mittal et al., 2012b) to deep neural network (DNN)-based methods (Talebi & Milanfar, 2018; Ying et al., 2020; Qin et al., 2023; Saha et al., 2023), bringing steady improvements in IQA accuracy and efficiency.

Nonetheless, these IQA methods still suffer from poor out-of-distribution (OOD) generalization ability (You et al., 2024) and expensive training costs (Wu

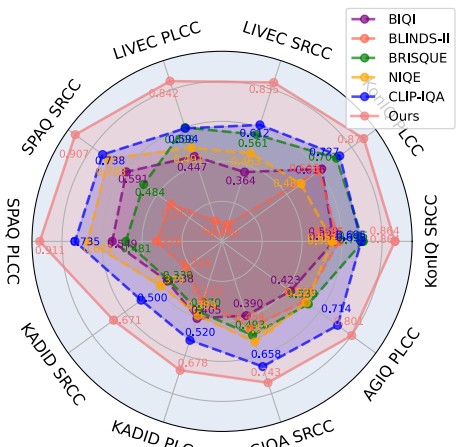

Figure 1: Comparison between LLM-IQA and existing training-free IQA SOTAs, exhibiting LLM-IQA's excellent zero-shot IQA ability.

et al., 2024a). One potential solution to the OOD issue involves training DNNs on a combination of multiple IQA datasets. Though sounds promising, this approach fails due to inconsistent standards used during dataset construction, leading to distribution mismatches across datasets. For instance, an image rated high quality in one dataset may receive a low-quality score in another, ultimately degrading model performance. Another approach is to create a larger, more diverse dataset representing a wide range of distortions. However, aside from the increased training costs, the scoring process is labor-intensive and time-consuming, making this approach impractical. As a result, poor OOD performance remains an open problem for current IQA area.

Figure 2: The idea of LLM-IQA is inspired by the human evaluator's scoring procedures. When scoring, human evaluators are provided with standards mapping the quality to scores. Then they start with the global quality and zoom in on objects to grasp local quality. We integrate these key procedures and switch their form according to MLLM's behavior pattern, formulating LLM-IQA.

Recently, MLLMs have shown impressive zero-shot capabilities across various computer vision tasks, including classification (Radford et al., 2021), segmentation (Li et al., 2024; He et al., 2024), detection (Zhang et al., 2023a), and restoration (Chen et al., 2023; Zhao et al., 2024). Thanks to their extensive training on large datasets and vast model sizes (Liu et al., 2024c; Awadalla et al., 2023), MLLMs possess rich prior knowledge and are closely aligned with human perceptual understanding (Yin et al., 2023). As the MLLM has not been trained on IQA-related datasets, previous related research (Wu et al., 2024a;c) mainly focuses on training or fine-tuning. These studies have demonstrated remarkable accuracy, suggesting that MLLMs hold great potential for driving the next wave of IQA advancements (see Figure 1). However, while fine-tuning significantly enhances accuracy, it introduces additional computational costs and complexity. Therefore, we aim to fully exploit MLLMs' potential **without resorting to fine-tuning or task-specific training**.

Our approach is inspired by the human evaluators' scoring process and the MLLMs' behavior pattern (Yin et al., 2023). Thus, we design an inference pipeline mimicking human image scoring which is shown in Figure 2. Our key designs stem from the following observations. **First**, when human score images, they are typically provided with a clear standard for each quality level (Wu et al., 2024b). Without such a standard, discrepancies arise—for example, one may interpret a score of 60 as merely passing, while another views 50 as average. By providing a consistent scoring standard, evaluators are more likely to agree on quality assessments. **Second**, when humans assess image quality, they consider both global and local quality (Navon, 1977; Gerlach & Poirel, 2018), often zooming in to evaluate specific areas (Förster, 2012). Notably, these zoomed-in evaluations are typically centered on objects within the image rather than being performed randomly. **Additionally**, MLLMs generate outputs in token form, making it difficult for them to produce precise scores, such as float number 86.5, which would require generating multiple tokens for 8, 6, dot and 5.

Building on these observations, we propose two novel techniques. **First**, we develop a standard-guided scoring system that aims to establish a clear mapping between quality levels and scores and restrict the MLLM to scoring within a predefined range $\{1, 2, \ldots, K\}$. The mapping and restriction ensure the model's understanding of the quality scale. **Second**, we utilize segmentation models to provide MLLM with the whole image and object-centered sub-images. We aggregate the scores using an area-weighted average approach. Our contributions can be summarized as follows:

- We propose **LLM-IQA**, a standard-guided mix-grained IQA framework that does not require any task-specific training or fine-tuning. LLM-IQA fully leverages the inherent capabilities of pre-trained MLLM and segmentation model to provide accurate IQA scores. Our LLM-IQA serves as a new paradigm for training-free approaches in IQA tasks.

- We design two key mechanisms to enhance performance. The standard-guided scoring mechanism ensures consistent and objective quality evaluation by aligning scores with predefined standards. The mix-grained aggregation mechanism refines the final quality score by aggregating global and object-centered sub-image quality scores.

- We conduct extensive experiments and compare LLM-IQA against SOTA IQA methods across multiple datasets. The main experiments show that our proposed LLM-IQA achieves SOTA performance compared with training-free methods, and competitive performance compared with training-based methods in cross-dataset scenarios.

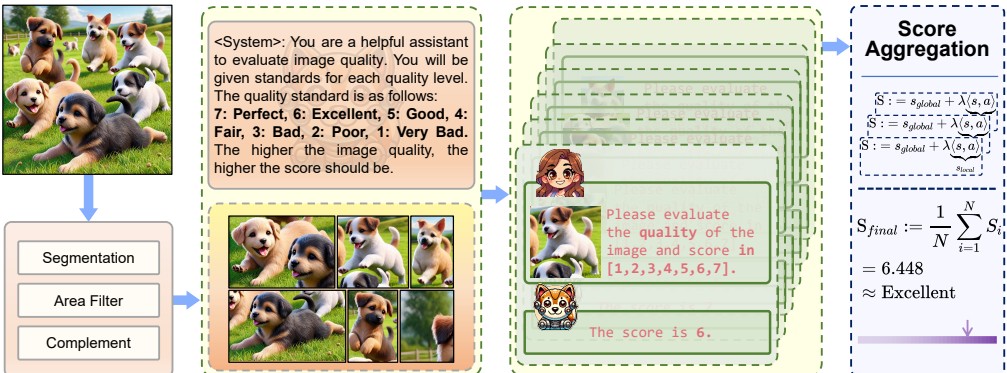

Figure 3: The overall pipeline for our proposed LLM-IQA. It can be divided into three stages, *i.e.,* segmentation, standard guided scoring, and score aggregation. The input image is segmented into multiple sub-images centered on objects. Then, MLLM scores with quality standards. After the area-weighted average, the scores from various models are aggregated as the final quality score, which falls in $[1, 7]$.

## 2 RELATED WORKS

**Training-Free IQA.** Training-free IQA is a critical approach in the field of image processing, allowing for the evaluation of image quality without the need for distortion-specific or human-rated training data. Traditional training-free IQA methods are often based on the statistical properties of images, focusing on full-reference (FR) metrics such as PSNR and SSIM (Wang et al., 2004). As for no-reference (NR) training-free IQA, NIQE (Mittal et al., 2012b) assesses image quality through the analysis of natural scene statistics features and provides robust but less precise results. In recent years, CLIP (Radford et al., 2021), a multimodal model, has emerged as a significant player, providing robust training-free performance support for prevalent deep-learning-based IQA. CLIP-IQA (Wang et al., 2023) explores the capabilities of CLIP for assessing image quality and aesthetic perception and pioneers the use of contrastive prompt strategies for scoring. ZEN-IQA (Miyata, 2024) and GRepQ (Srinath et al., 2024) also harness CLIP, with ZEN-IQA utilizing antonym prompts and GRepQ combining low-level and high-level feature representations for IQA. While these inspiring developments represent a substantial leap forward, there is still huge potential for enhancing the performance of training-free IQA models in accuracy.

**MLLMs for IQA.** High-performance MLLMs, such as mPLUG-Owl (Ye et al., 2023; 2024b;a), LLaVA (Liu et al., 2024c;a;b), and InternLM-XComposer (Zhang et al., 2023b; Dong et al., 2024), can be exceptionally utilized to align IQA tasks with human perception. Based on a comprehensive study (Wu et al., 2024d), recent efforts concentrate on benchmarking and fine-tuning MLLMs for IQA. Q-Bench (Wu et al., 2023) and DepictQA (You et al., 2024) establish evaluation benchmarks for the perceptual, descriptive, comparative, and evaluative capabilities of MLLMs in low-level vision. Based on these works, Q-Instruct (Wu et al., 2024a) and Co-Instruct (Wu et al., 2024c) further advance the low-level perceptual and descriptive capabilities of MLLMs by introducing large-scale datasets and conducting pre-training. Q-Align (Wu et al., 2024b) categorizes image quality into five tiers, enabling more precise quality score regression. However, the cost of fine-tuning large models is substantial, prompting the consideration of more efficient approaches.

## 3 METHODOLOGY

We provide a comprehensive explanation of our proposed LLM-IQA method. The overall pipeline of our proposed LLM-IQA is shown in Figure 3. The input image is segmented into multiple sub-images with the segmentation process pipeline. Given a detailed standard, the MLLM rates the whole image and sub-images with scores in $\{1, 2, \ldots, 7\}$. These scores will be finally aggregated to form the final number. Specifically, we first propose the standard-guided scoring mechanism, which effectively leverages its prior knowledge and its behavior pattern. Second, we discuss the mix-grained aggregation mechanism, which consists of the process of obtaining suitable sub-images and the aggregation of scores. The rationale behind using segmentation is also included.

## 3.1 STANDARD-GUIDED SCORING MECHANISM

The ultimate goal of image quality assessment (IQA) is to evaluate images in a manner that closely mirrors human judgment. Thanks to their extensive training data and vast prior knowledge, MLLMs are capable of perceiving images in a way that aligns with human perception (Wu et al., 2023), giving them an inherent advantage for IQA tasks. However, expecting an MLLM to output precise quality scores, such as 87.5, is impractical. This is because a score like 87.5 is not represented by a single token, but by four separate tokens: 8, 7, dot, and 5 respectively. Typically, MLLMs can hardly grasp the internal relationship between these tokens, making it difficult for them to associate these values with image quality. These observations and analyses motivate us to **insight 1**:

*It is more effective to represent image quality using one single token to achieve an accurate score.*

Additionally, relying solely on numeric outputs may not be the most optimal approach for two key reasons. First, numbers constitute only a small fraction of the data within the training set compared to text. However, using only text is also not feasible, as we still need to extract a quantitative score. Second, human interpretation of numeric scores can vary. For instance, some may consider a score of 60 to be just passing, while others may view 50 as an average score. Therefore, when human evaluators score image quality, they are often provided with clear standards for each level of quality (Wu et al., 2024b). This observation brings us to **insight 2**:

*A combination of text and numbers is a more effective prompt format for MLLM IQA.*

In our proposed method, we integrate these two insights and design the prompt for LLM-IQA as follows:

> *# System:You are a helpful assistant to evaluate image quality. You will be given standards for each quality level. The quality standard is listed as follows: 7: Perfect, 6: Excellent, 5: Good, 4: Fair, 3: Bad, 2: Poor, 1: Very Bad. The higher the image quality is, the higher the score should be.*
> *# User:* `` *Please evaluate the quality of the image and score in [1, 2, 3, 4, 5, 6, 7].*

In LLM-IQA, MLLM only outputs discrete numbers from 1 to 7. While this discrete scoring approach may introduce a slight loss in precision compared to continuous values, the impact is minimal. Denote that integer score as $s \in \{s|s \in \mathbb{Z}^+ \wedge 1 \leq s \leq K\}$, the ground truth MOS as $s^*$, and the maximal and minimal value of $s^*$ as $\text{Max}_{gt}$ and $\text{Min}_{gt}$ respectively. We scale $s^*$ to $\{0, 1, \ldots, K-1\}$ and round it to the nearest integer. The conversion formula is written as:

Table 1: The approximation of the $(SRCC + PLCC)/2$ upper bound of using only K integers to score.

| K | SPAQ | KonIQ | LIVEC | AGIQA | KADID |
|---|------|-------|-------|-------|-------|
| 3 | 0.912 | 0.830 | 0.915 | 0.923 | 0.942 |
| 5 | 0.968 | 0.946 | 0.964 | 0.973 | 0.980 |
| 7 | 0.983 | 0.967 | 0.982 | 0.986 | 0.988 |
| 9 | 0.990 | 0.979 | 0.989 | 0.991 | 0.993 |

$$\hat{s}^* = \text{Round}(K(s^* - \text{Min}_{gt})/(\text{Max}_{gt} - \text{Min}_{gt})). \tag{1}$$

As shown in Table 1, the performance upper bounds for different $K$ demonstrate that even when using a limited number of discrete levels, the results surpass those of existing methods. The precision loss introduced by the conversion to discrete scores is minimal and can be safely ignored. However, it is not the case that greater K brings better performance when considering MLLM and there is a performance turning point. We will analyze this problem in Sec 4.4.

In conclusion, for each image $X$, MLLM processes its corresponding segmented masks $M$ as input. For each mask $m_k \in M$, MLLM will predict a score $s_k \in \{1, 2, \ldots, K\}$. These individual scores form a score list $\boldsymbol{s}_i$, which is subsequently used to compute the final quality score.

## 3.2 MIX-GRAINED AGGREGATION MECHANISM

The mix-grained aggregation mechanism can be divided into two parts. The first part introduces the segmentation pipeline, while the other presents the aggregation of multiple scores.

**Segmentation Process Pipeline.** When humans recognize an image, they start from the global structure and gradually dive into the local parts (Navon, 1977; Förster, 2012; Gerlach & Poirel, 2018). This hierarchical process also applies when assessing image quality. Therefore, under the assumption that MLLMs share a similar perception process, it is essential to leverage meaningful sub-images

Table 2: Performance comparison of LLM-IQA with other **training-free** IQA models on KonIQ, LIVE Challenge, SPAQ, KADID-10k and AGIQA-3k. **Bold** font indicates the best performance.

| Methods | KonIQ | | LIVE Challenge | | SPAQ | | KADID-10k | | AGIQA-3k | |
|---|---|---|---|---|---|---|---|---|---|---|
| | SRCC ↑ | PLCC ↑ | SRCC ↑ | PLCC ↑ | SRCC ↑ | PLCC ↑ | SRCC ↑ | PLCC ↑ | SRCC ↑ | PLCC ↑ |
| BIQI (Moorthy & Bovik, 2010) | 0.559 | 0.616 | 0.364 | 0.447 | 0.591 | 0.549 | 0.338 | 0.405 | 0.390 | 0.423 |
| BLIINDS-II (Saad et al., 2010) | 0.585 | 0.598 | 0.090 | 0.107 | 0.317 | 0.326 | 0.224 | 0.313 | 0.454 | 0.510 |
| BRISQUE (Mittal et al., 2012a) | 0.705 | 0.707 | 0.561 | 0.598 | 0.484 | 0.481 | 0.330 | 0.370 | 0.493 | 0.533 |
| NIQE (Mittal et al., 2012b) | 0.551 | 0.488 | 0.463 | 0.491 | 0.703 | 0.671 | 0.379 | 0.389 | 0.529 | 0.520 |
| CLIP-IQA (Wang et al., 2023) | 0.695 | 0.727 | 0.612 | 0.594 | 0.738 | 0.735 | 0.500 | 0.520 | 0.658 | 0.714 |
| **LLM-IQA (Ours)** | **0.864** | **0.875** | **0.835** | **0.842** | **0.907** | **0.911** | **0.671** | **0.678** | **0.743** | **0.801** |

deliberately. Specifically, 'meaningful' means that these sub-images should not be obtained through random cropping but through semantic segmentation techniques.

The segmentation model is an excellent choice as it tends to segment the semantic objects out. The object segmented by the segmentation model is padded with zeros around. While this padding has minimal impact on human perception, as humans can easily recognize the black padding as meaningless and mentally disregard it, this is not the case for MLLMs. The visual encoder within the MLLM processes the padding as part of the actual image, leading the model to misinterpret the black regions as the real background. This misunderstanding can result in distinct errors, such as the MLLM perceiving low contrast when the foreground is dark or concluding that the background is too dark. Both cases can negatively affect image quality assessment's accuracy.

To address the above issue, we adopt an alternative approach by padding the segmented areas with the original pixel values. Besides, the segmented results of most segmentation models are highly fine-grained. However, small objects tend to have lower image quality due to insufficient pixel density, making it difficult to display sharp details. To mitigate, we apply a coarser granularity and establish a minimum threshold $t$ for mask size. A side effect of coarser granularity is that the masks may only cover a portion of the image. In some extreme cases, the segmentation model may fail to segment any objects from low-quality images. To compensate for this problem, we create a new mask for the uncovered portions of all previous masks. The detailed process of the segmentation pipeline is in Algorithm 1.

**Assessment Score Aggregation.** For a given image $X_i$, after obtaining its global score $s_{global}$, segmented masks $M_i$, and their corresponding scores $s_i$, we proceed to compute the final predicted score. A simple approach of averaging $s$ to determine the final score for $X_i$ yields poor performance. This is because some blurred objects are too small to be perceptible to humans but are disproportionately penalized by MLLMs, leading to an unfairly low score.

To address this, we propose a weighted average of the scores, where the area of the corresponding masks determines the weights. Mathematically, this can be written as $s_{local} = <s, a>$, where $< \cdot, \cdot >$ is inner product and $a$ is the vector of the areas of the masks in $M$. This approach aligns more closely with human perception, as the

---

**Algorithm 1:** Segmentation Process Pipeline

**Data:** Dataset $\mathcal{D} = \{X_i\}_{i=1}^N$, area threshold $t$, pretrained SAM2 $\mathcal{S}$

**Result:** Masks $\mathcal{M} = \{M_i\}_{i=1}^N$

$\mathcal{M} \leftarrow []$;

**foreach** *image $X_i$ in D* **do**
  $raw\_masks \leftarrow \mathcal{S}(X_i)$, $final\_masks \leftarrow []$;
  **foreach** *mask m in raw_masks* **do**
    **if** *m.area $\geq t$* **then**
      $final\_masks.append(mask)$;
    **end**
  **end**
  $remain\_mask = \bigcap\{\neg final\_masks\}$;
  **if** *remain_mask.area $\geq t$* **then**
    $final\_masks.append(remain\_mask)$;
  **end**
  $\mathcal{M}.append(final\_masks)$;
**end**
**return** $\mathcal{M}$;

---

dominant object in an image typically occupies the largest region, which represents the image's quality. Therefore, for one MLLM $k$, the score is given by $s_k = s_{global} + \lambda s_{local}$. Consequently, we apply model ensemble to aggregate scores from various MLLMs by $s_{Dog} = \sum_{k=1}^{N_{model}} s_k / N_{model}$, where $N_{model}$ is the number of MLLMs.

## 4 EXPERIMENTS

### 4.1 EXPERIMENTAL SETTINGS

**Datasets.** We select the following datasets to evaluate our IQA method: KonIQ (Hosu et al., 2020), LIVE Challenge (Ghadiyaram & Bovik, 2015), SPAQ (Fang et al., 2020), KADID (Lin et al., 2019),

and AGIQA (Li et al., 2023). KonIQ and SPAQ are large in-the-wild IQA datasets with more than 10k images. LIVE Challenge is a smaller in-the-wild dataset with 1.1k images. KADID-10k is a large synthetic dataset, while AGIQA-3k focuses on AI-generated images. Together, these datasets provide a comprehensive range of image types and quality variations for accurate model evaluation.

**State-of-the-art Methods.** We compare our training-free LLM-IQA's performance against two categories of approaches. The first category is training-free methods, including BIQI (Moorthy & Bovik, 2010), BLINDS-II (Saad et al., 2010), BRISQUE (Mittal et al., 2012a), NIQE (Mittal et al., 2012b), and CLIP-IQA (Wang et al., 2023). The second is training-based methods such as NIMA (Talebi & Milanfar, 2018), DBCNN (Zhang et al., 2020), HyperIQA (Su et al., 2020), MUSIQ (Ke et al., 2021), CLIP-IQA+ (Wang et al., 2023), LIQE (Zhang et al., 2023c), and Q-Align (Wu et al., 2024b). CLIP-IQA and Q-Align are currently the SOTA IQA models without and with training respectively.

**Evaluation.** All methods are evaluated in cross-dataset scenarios to demonstrate their zero-shot capabilities. Comparing training-free methods with training-based methods may seem unfair due to the latter's systematic training on quality assessment. We still perform these comparisons to showcase the robustness and competitive zero-shot performance of our approach. The evaluation metrics used are Spearman's rank correlation coefficient (SRCC) and Pearson's linear correlation coefficient (PLCC). Both metrics are widely used in IQA to assess the correlation between the model's predictions and human judgments, typically represented by MOS (Telecom, 2000).

**Implementation Details.** We select the pre-trained SAM2 (Ravi et al., 2024) as the segmentation model and mPLUG-Owl3 (Ye et al., 2024a) as the MLLM. The hyperparameters of SAM2 are adjusted to achieve the desired segmentation granularity, with detailed configurations in the supplementary material. Using these hyperparameters, the average number of masks generated for the SPAQ dataset is 7.22. For mPLUG-Owl3, we limit the output length to 1 token and utilize its default hyperparameters across all test sets. The number of standard words is set as $K = 7$. Our code is written with PyTorch (Paszke et al., 2019) and runs with NVIDIA RTX A6000 GPU.

## 4.2 Comparison with State-of-the-art Methods

We conduct extensive experiments to evaluate the performance of our proposed LLM-IQA model. The comparisons with SOTA methods are divided into two categories: training-free methods, shown in Table 2, and training-based methods, as presented in Table 5. Both comparisons highlight our excellent performance on zero-shot IQA.

**Training-Free methods.** Training-free methods can be broadly categorized into two types. The first category includes CLIP-IQA, which leverages the prior knowledge of CLIP and generates scores based on the similarity between text and image embeddings. The second category consists of models such as BIQI, BLINDS-II, BRISQUE, and NIQE, which rely on hand-crafted features. As shown in Table 2, the traditional hand-crafted features often fail to score accurately due to the complex nature of human opinions on image quality. CLIP-IQA benefits from its prior knowledge and demonstrates higher accuracy than hand-crafted feature-based methods. Our LLM-IQA

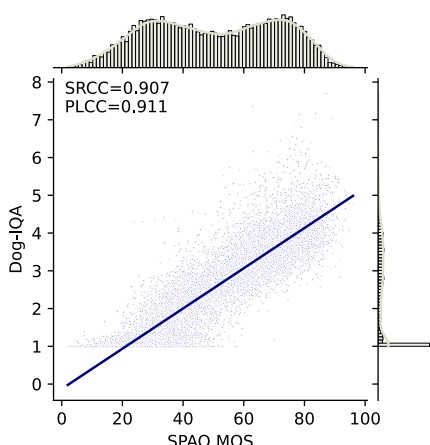

Figure 4: Correlation between MOS and LLM-IQA's scores on SPAQ. The points $(s^*, s_{Dog})$ are scattered in the center. And the marginal hist plots show the distribution of GT and LLM-IQA's scores.

model consistently achieves superior performance across all metrics and datasets, significantly outperforming existing training-free methods.

**Training-Based methods.** Table 5 summarizes the performance of various training-based methods in cross-dataset evaluations. These experiments test the OOD generalization ability of the models, which is crucial for IQA. For these comparisons, we select KonIQ and SPAQ as training sets due to their large size and in-the-wild characteristics. Notably, our LLM-IQA method requires **no training or fine-tuning**, making its strong performance even more remarkable.

Figure 5: Performance comparison of our model with **training-based** IQA models. The best and second-best performance is indicated by **bold** and underlined respectively.

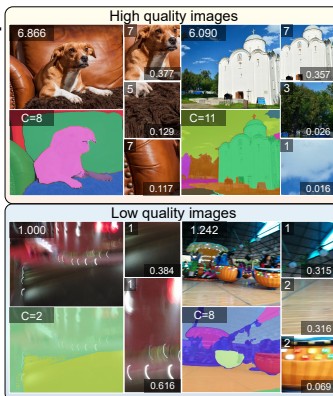

| Training: KonIQ | →Testing Set: | SPAQ | | AGIQA-3k | | KADID-10k | |
|---|---|---|---|---|---|---|---|
| Method | Training-free? | SRCC ↑ | PLCC ↑ | SRCC ↑ | PLCC ↑ | SRCC ↑ | PLCC ↑ |
| NIMA | × | 0.856 | 0.838 | 0.654 | 0.715 | 0.535 | 0.532 |
| DBCNN | × | 0.806 | 0.812 | 0.641 | 0.730 | 0.484 | 0.497 |
| HyperIQA | × | 0.788 | 0.791 | 0.640 | 0.702 | 0.468 | 0.506 |
| MUSIQ | × | 0.863 | 0.868 | 0.630 | 0.722 | 0.556 | 0.575 |
| CLIP-IQA+ | × | 0.864 | 0.866 | 0.685 | 0.736 | 0.654 | 0.653 |
| LIQE | × | 0.833 | 0.846 | 0.708 | 0.772 | 0.662 | 0.667 |
| Q-Align | × | 0.887 | 0.886 | 0.735 | 0.772 | **0.684** | **0.671** |
| **LLM-IQA** (Ours) | ✓ | **0.907** | **0.911** | **0.743** | **0.801** | 0.671 | 0.678 |

| Training: SPAQ | →Testing Set: | KonIQ | | AGIQA-3k | | KADID-10k | |
|---|---|---|---|---|---|---|---|
| Method | Training-free? | SRCC ↑ | PLCC ↑ | SRCC ↑ | PLCC ↑ | SRCC ↑ | PLCC ↑ |
| NIMA | × | 0.733 | 0.788 | 0.534 | 0.630 | 0.399 | 0.480 |
| DBCNN | × | 0.731 | 0.758 | 0.459 | 0.518 | 0.490 | 0.508 |
| HyperIQA | × | 0.714 | 0.742 | 0.570 | 0.649 | 0.381 | 0.448 |
| MUSIQ | × | 0.753 | 0.680 | 0.564 | 0.675 | 0.349 | 0.429 |
| CLIP-IQA+ | × | 0.753 | 0.777 | 0.577 | 0.614 | 0.633 | 0.638 |
| LIQE | × | 0.826 | 0.847 | 0.672 | 0.772 | 0.639 | 0.627 |
| Q-Align | × | 0.848 | **0.879** | 0.723 | 0.786 | **0.743** | **0.740** |
| **LLM-IQA** (Ours) | ✓ | **0.864** | 0.875 | **0.743** | **0.801** | 0.671 | 0.678 |

Figure 6: Examples are selected to present **LLM-IQA**'s ability. Scores are on the upper left, and the area on the lower right. Zooming in for a better view.

Table 3: Ablation study of our proposed LLM-IQA on SPAQ, AGIQA-3k, and LIVE Challenge. We test the influence of the aggregation method, segmentation method, and standard given to MLLM. Our key designs are significant in improving MLLM scoring accuracy.

| | Settings | | | SPAQ | | AGIQA-3k | | LIVE Challenge | | Average |
|---|---|---|---|---|---|---|---|---|---|---|
| Exp index | Aggregation | Segmentation | Standard | SRCC ↑ | PLCC ↑ | SRCC ↑ | PLCC ↑ | SRCC ↑ | PLCC ↑ | |
| 1 | N/A | Whole | Without | 0.616 | 0.649 | 0.693 | 0.752 | 0.663 | 0.651 | 0.671 |
| 2 | Area | BBox | Number | 0.680 | 0.624 | 0.443 | 0.464 | 0.349 | 0.319 | 0.480 |
| 3 | Area | BBox | Sentence | 0.793 | 0.761 | 0.592 | 0.615 | 0.619 | 0.598 | 0.663 |
| 4 | Mean | BBox | Word | 0.866 | 0.831 | 0.652 | 0.624 | 0.687 | 0.638 | 0.716 |
| 5 | Area | Mask | Word | 0.632 | 0.540 | 0.510 | 0.431 | 0.454 | 0.395 | 0.494 |
| 6 | N/A | Whole | Word | 0.891 | 0.883 | 0.680 | 0.684 | 0.708 | 0.707 | 0.759 |
| 7 | Area | BBox | Word | 0.851 | 0.792 | 0.659 | 0.642 | 0.667 | 0.628 | 0.707 |
| 8 | Area | BBox+Whole | Word | 0.886 | 0.885 | 0.687 | 0.689 | 0.739 | 0.718 | 0.767 |
| 9 | Area+Ensemble | BBox+Whole | Word | **0.907** | **0.911** | **0.703** | **0.801** | **0.835** | **0.842** | **0.833** |

Training-based methods show variability depending on the dataset used for training. For example, SRCC and PLCC scores of Q-Aling on KADID-10k drop significantly when switching the training set from KonIQ to SPAQ, despite both being in-the-wild datasets. In contrast, LLM-IQA demonstrates stable performance without any training, highlighting its advantage in terms of generalization and cost-efficiency. Moreover, scoring AI-generated images has become increasingly critical in the current era of AI. LLM-IQA gains the best performance on AGIQA-3k dataset, exhibiting its superiority on AI-generated content. Moreover, in the KonIQ → SPAQ scenario, LLM-IQA achieves the highest SRCC (0.907) and PLCC (0.911), clearly outperforming the second-best model, which only achieves 0.887 SRCC. This result underscores the superiority of LLM-IQA in cross-dataset evaluations. However, LLM-IQA's performance is relatively low in KADID-10k. This may be due to the fact that KADID is a synthetic dataset and its distribution is significantly different from other datasets. Despite this, LLM-IQA still secures the second-best performance, highlighting its robustness. In conclusion, LLM-IQA's consistently ranks at the top or near the top positions across all datasets, which demonstrates its robustness and effectiveness as a training-free IQA approach.

### 4.3 VISUALIZATOIN

**Example Images.** Figure 6 shows example segmentations and scoring results. We have selected both high-quality and low-quality images. For each image, the following figures are provided: the full image, the segmented image, and three exemplary masks. Then we will introduce the meaning of the numbers around the corners of each image. The upper left corner of the full image displays the final score predicted by our LLM-IQA model. Directly below the full image, the segmentation results are shown, with the mask count indicated in the upper left. On the right, three masks of varying quality are presented. Furthermore, each mask is annotated with their corresponding scores (upper left) and area weights (lower right). From these example figures, we can directly perceive the model's segmentation results and thus come to the following conclusions.

By incorporating image segmentation, MLLM is capable of capturing local distortions within the object-centered images. This allows assigning scores to different regions that correspond to their quality, rather than relying on a single overall score. This enables MLLM to achieve human-aligned quality perception. In conclusion, LLM-IQA provides accurate scores for different quality levels.

**Score Distribution.** We also visualize the scores predicted by humans and our proposed LLM-IQA on SPAQ datasets in Figure 4. The range of the final score varies between 1 and 7 and most of the scores are not integers. This is because the final score consists of the area-weighted average of scores and the number of masks. As the scores from MLLM are discrete, the final scores are denser around the integer values. The area average mechanism helps the continuous-like distribution.

## 4.4 Ablation Study

The ablation studies provided in Tables 3, 4 highlight the significance of various components in our proposed LLM-IQA model. By systematically altering key aspects of the model, the experiment evaluates how each component affects performance on two datasets: SPAQ and AGIQA-3k. We examine components including 1) the number of tokens, 2) the standard given to MLLM, 3) the selection of the mask and bounding box, 4) the aggregation method, 5) the influence of global and local quality, 6) the number of words, and 7) the impact of segmentation pipeline. The experiment results are shown in Tables 3, 4, and 5. Next, we will analyze the influence of each component.

**Single Token.** Experiment 1 in Table 3 asks MLLM to output float numbers to judge the quality of images, while experiment 6 takes one token for each image. This pair of experiments shows the significant improvement when limiting the output of MLLM to a single token. MLLM usually performs poorly when describing float numbers with multiple tokens. This pair strongly supports our insight 1, which says it is more effective to represent image quality using one single token to achieve an accurate score.

**Standard.** Standard-guided scoring is a critical aspect of LLM-IQA. We compare three forms of standards, namely number, word, and sentence. The number standard asks the MLLM to rate image quality in the range $\{1, 2, \ldots, K\}$. The word standard adds descriptive adjectives, such as *excellent* and *bad*, to each score. The sentence standard assigns a sentence describing quality.

As shown in experiments 2, 3, and 7 in Table 3, the word-based standard yields the best performance as it provides an accurate mapping between number and quality. While sentences offer more detailed context than numbers, they can introduce abstract terms (e.g., *some*, *certain*) that may distract the model, resulting in slightly lower performance. Numbers, on the other hand, perform poorly because the MLLM struggles to understand their relationship to image quality without additional context. In conclusion, associating a word with each score effectively enhances accuracy.

**Mask and Bounding Box.** When scoring sub-images, we test three kinds of input formats: masks (semantic object coverings), bounding boxes (enclosing the masks), and the entire image. As shown in experiments 5, 6, and 7 in Table 3, using masks significantly degrades performance. This is mainly because the constant padding applied to masked areas is still interpreted by the MLLM's visual encoder, negatively influencing the score. Conversely, using the entire image as input provides moderate results, though still inferior to bounding boxes. Bounding boxes improve performance without computational overhead as the padding is always calculated by the visual encoder and adds no more tokens for LLM. Therefore, applying bounding boxes as a segmentation method is necessary for maximizing LLM-IQA's accuracy.

**Score Aggregation.** We evaluate two score aggregation methods: simple average and area-weighted average. Considering that the summation of the area should be the area of the image, we use the mask area instead of the sub-image area. As experiments 4 and 7 in Table 3 indicate, there is a significant improvement in both datasets with area-weighted average. This can be explained by the attention scheme. There are plenty of small objects that are often scored with low quality because of a lack of pixels. However, the quality of the image is always represented by the main object, which usually has a larger area. So more attention should be put on larger objects, namely taking the area-weighted average on quality scores of sub-images, which is more consistent with humans. In conclusion, leveraging the area-weighted average effectively improves LLM-IQA's accuracy.

**Global and Local Quality.** To validate the significance of local quality versus global quality, we conduct experiments 6, 7, and 8, with results presented in Table 3. From the experimental results, we can draw two critical conclusions. **First**, the overall information exceeds the sum of the quality information from various local sources. Global quality gains higher SRCC (0.891) on SPAQ than

local quality (0.851). This observation highlights the effectiveness of our fine-grained evaluation methodology and the innovative design of our score aggregation process. **Second**, although neither the local scores nor the overall score reaches 0.9 SRCC, the summation of the two can still further enhance the model's accuracy. For simplicity, we take the summation of global and local scores. In summary, the experimental results strongly support the notion that the integration of both global and local quality, namely mix-grained, yields superior results compared to the isolated performance of each.

**Number of Words.** As discussed before, after applying a discrete scoring form, the number of levels decides the performance upper bound of IQA models. So we test the performance of our proposed LLM-IQA with 3, 5, 7, and 9 words. All numbers are odd because there needs to be a level representing medium to conform

Table 4: Number of words (K).

| K | SPAQ | | KADID-10k | | AGIQA-3k | | Average |
|---|---|---|---|---|---|---|---|
| | SRCC ↑ | PLCC ↑ | SRCC ↑ | PLCC ↑ | SRCC ↑ | PLCC ↑ | |
| 3 | 0.731 | 0.722 | 0.447 | 0.473 | 0.747 | 0.757 | 0.646 |
| 5 | 0.853 | 0.860 | 0.572 | 0.576 | **0.808** | **0.797** | 0.744 |
| 7 | **0.885** | **0.875** | 0.580 | **0.589** | 0.800 | 0.779 | **0.751** |
| 9 | 0.875 | 0.840 | **0.583** | 0.586 | 0.743 | 0.753 | 0.730 |

to human evaluation. The result is shown in Table 4. Only three words are not enough to gain excellent performance while it still surpasses most of the previous training-free methods (see Table 2). Interestingly, the results also indicate that increasing the number of levels beyond a certain point does not necessarily lead to better performance. Specifically, using 7 words yields the best results in most scenarios and the second-best in the remaining cases. In summary, 7 appears to be the optimal number of word levels to accurately assess image quality.

**Segmentation.** In Table 5, we present the performance of LLM-IQA with default setting, a much smaller SAM, and no complement mask. The default setting provides segmented masks with too fine granularity, which results in lower performance and much longer inference time. The tiny SAM version leads to coarser segmentation around the boundary.

Table 5: Segmentation Setting.

| Setting | SPAQ | | AGIQA-3k | | LIVE Challenge | | Average |
|---|---|---|---|---|---|---|---|
| | SRCC ↑ | PLCC ↑ | SRCC ↑ | PLCC ↑ | SRCC ↑ | PLCC ↑ | |
| Default | 0.875 | 0.883 | 0.686 | **0.689** | 0.706 | **0.718** | 0.760 |
| SAM Tiny | 0.879 | 0.884 | **0.687** | 0.685 | **0.742** | 0.717 | 0.766 |
| No Complement | 0.675 | 0.637 | 0.488 | 0.488 | 0.425 | 0.428 | 0.524 |
| Our Setting | **0.886** | **0.885** | **0.687** | **0.689** | 0.739 | **0.718** | **0.767** |

However, as we leverage the bounding box to fill the mask, the boundary is not that important. If we remove the complement part in the segmentation pipeline, the results degrade a lot. Therefore, we adjust SAM parameters to reduce the number of masks and its impact on performance is minor. However, the design of the complement mask is necessary to achieve accurate IQA.

## 5 LIMITATIONS AND DISCUSSIONS

**MLLM Inference Speed.** Because the MLLM must evaluate the quality of each mask, the inference speed of LLM-IQA is relatively slow compared to models that require only a single inference. On average, LLM-IQA processes 7.22 masks and the entire image, resulting in $7\times$ longer inference time. After testing on a single NVIDIA RTX A6000 GPU, our proposed LLM-IQA can segment the whole SPAQ dataset with 10k images in 30 minutes and score each mask and the total data within 8 hours. The resolution of most images are around and above $1024 \times 1024$. This process can be performed with data parallel, which means it takes around 1.5 hours to obtain the final result when running on 4 GPUs. While the text embeddings can be pre-calculated and reused, allowing for the omission of the text encoder, the total inference time remains longer than single forward inference.

## 6 CONCLUSION

In this work, we propose LLM-IQA, a standard-guided zero-shot mix-grained IQA method, which is training-free and utilizes the exceptional prior knowledge of MLLMs. With the combination of SAM2 and mPLUG-Owl3, we propose two key mechanisms to enhance IQA performance. The standard-guided scoring mechanism ensures consistent and objective quality evaluation by aligning scores with predefined standards. The mix-grained aggregation mechanism refines the final quality score by aggregating global and object-centered sub-image quality scores. We conduct extensive experiments across a variety of datasets, benchmarking our proposed LLM-IQA against SOTA methods. The results demonstrate that LLM-IQA outperforms all previous training-free approaches and achieves competitive performance relative to training-based methods, which strongly supports the novelty and robustness of our proposed mechanisms. Future research will target reducing the computational costs associated with inferences and enhancing pixel-level quality assessments.

## A    ETHICS STATEMENT

The research conducted in the paper conforms, in every respect, with the ICLR Code of Ethics.

## B    REPRODUCIBILITY STATEMENT

We have provided implementation details in Sec. 4. We will also release all the code and models.

## C    LLM USAGE STATEMENT

Large Language Models (LLMs) were used solely for polishing writing. They did not contribute to the research content or scientific findings of this work.

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
