# LLM-IQA: Standard-guided MLLM for Mix-grained Image Quality Assessment

## A    Metrics

We employed Spearman's rank correlation coefficient (SRCC) and Pearson's linear correlation coefficient (PLCC) as the evaluation metrics. Both metrics fall within the range of $[-1, 1]$, and the performance is considered better when they have higher absolute values. Here, we provide the calculation formula for both metrics.

**Pearson linear correlation coefficient formula.** For two variables $X$ and $Y$, assuming there are $n$ pairs of observations $(x_1, y_1), (x_2, y_2), \cdots, (x_n, y_n)$, the Pearson correlation coefficient $r$ is calculated as:

$$r = \frac{\sum_{i=1}^{n}(x_i - \overline{x})(y_i - \overline{y})}{\sqrt{\sum_{i=1}^{n}(x_i - \overline{x})^2}\sqrt{\sum_{i=1}^{n}(y_i - \overline{y})^2}}, \tag{1}$$

where $\overline{x}$ is the mean of $X$ and $\overline{y}$ is the mean of $Y$.

**Spearman rank-order correlation coefficient formula.** First, rank the observations of $X$ and $Y$ respectively to obtain the corresponding ranks $R(X)$ and $R(Y)$. Assuming there are $n$ pairs of observations, the Spearman rank correlation coefficient $\rho$ is calculated as:

$$\rho = 1 - \frac{6\sum_{i=1}^{n} d_i^2}{n(n^2 - 1)}, \tag{2}$$

where $d_i = R(X_i) - R(Y_i)$, representing the rank difference between $X$ and $Y$ at the $i$-th observation.

In conclusion, PLCC focuses on the linear correlation between two variables while SRCC focuses on the rank of each element in its variables.

## B    Segmentation Settings

The parameters for the SAM2 (Ravi et al., 2024) automatic mask generator are listed in Table 2. As for parameters not mentioned, we use the default parameters. These parameters are selected to obtain the desired granularity. Even though, many objects are too small or with irregular shapes.

With these parameters, the distribution of masks is shown in Figure 1.

## C    Performance on other MLLM

We evaluate the impact of standard-guided mechanism on different models and the result is shown in Table 3. Both mPLUG-Owl3 and Janus-Pro-7B models' performance on SPAQ without LLM-IQA is relatively low. With LLM-IQA, the SRCC scores increase 0.270 and 0.295 for mPLUG-Owl3 and Janus-Pro-7B respectively.

## D    Visualization

In Figure 2, we visualize more example segmentations and scoring results. Our examples are drawn from diverse datasets, including natural scenes from SPAQ (Fang et al., 2020), LIVEC (Ghadiyaram

Table 1: The approximation of performance upper bound of using only K integers to score. The value is calculated by $(SRCC + PLCC)/2$.

Table 2: Parameters of SAM2

| Variable | Value |
| --- | --- |
| points_per_side | 8 |
| points_per_batch | 128 |
| pred_iou_thresh | 0.9 |
| stability_score_thresh | 0.8 |
| stability_score_offset | 0.7 |
| crop_n_layers | 0 |
| box_nms_thresh | 0.9 |
| crop_n_points_downscale_factor | 1.2 |
| min_mask_region_area | 1000 |
| use_m2m | False |
| output_mode | 'coco_rle' |

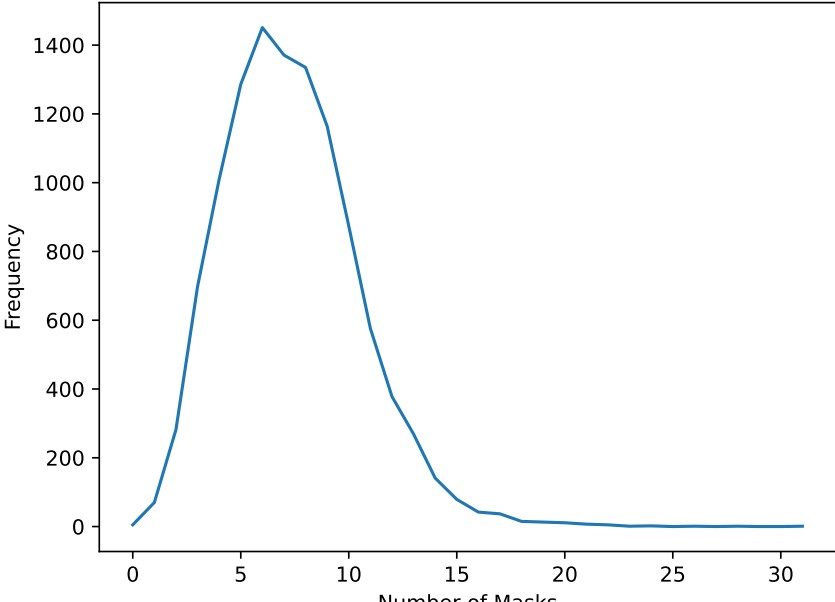

Figure 1: The distribution of numbers of masks with our SAM2 parameters on SPAQ.

& Bovik, 2015), and KonIQ (Hosu et al., 2020), synthetic images with distortions from KADID (Lin et al., 2019), and AI-generated images from AGIQA (Li et al., 2023). These datasets offer a diverse range of sources for IQA tasks, and can effectively evaluate the performance of our model across different IQA challenges.

For clarity, our model's evaluation results are organized into five descending score categories. And for each category, we display images along with their masks, and select additional images with corresponding scores. The score is indicated by the number in the upper left corner, while the area is

| SRCC/PLCC | mPLUG-Owl3 | Janus-Pro-7B |
| --- | --- | --- |
| w/o LLM-IQA | 0.616/0.649 | 0.362/0.310 |
| w LLM-IQA | 0.886/0.885 | 0.657/0.605 |

Table 3: Performance of other model with and without LLM-IQA on SPAQ.

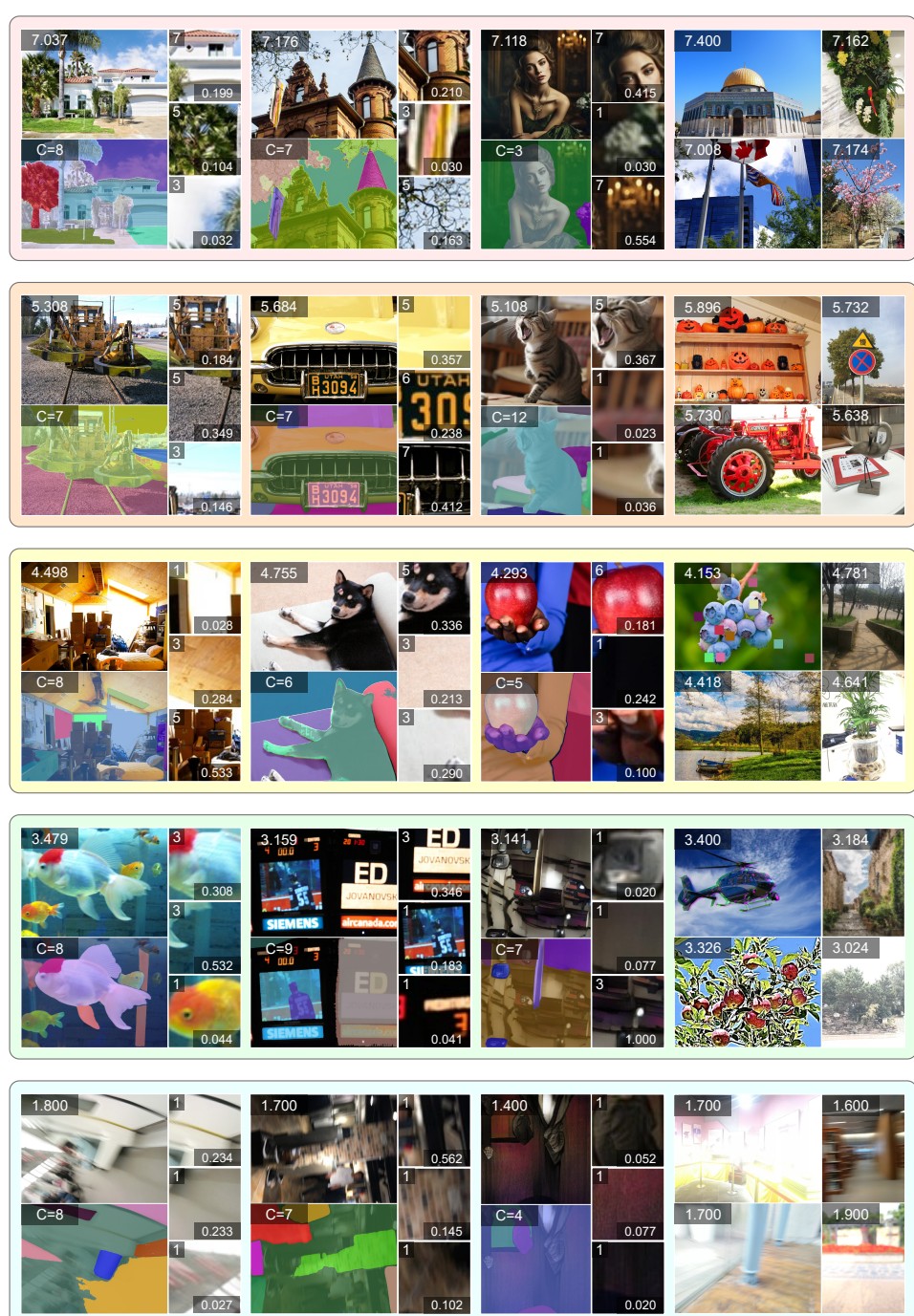

Figure 2: More example images with their segmented images and the LLM-IQA predicted scores. The results show that LLM-IQA can detect local distortions and accurately score images like human.

noted in the lower right corner. The number of masks is shown in the upper left part in the segmented image.

From these examples, we can further obtain conclusions as follows. **First**, our model, LLM-IQA, possesses the ability to precisely categorize images into distinct quality levels, with clear score differences between each level. It consistently aligns with human assessments across all levels, demonstrating its reliability in IQA tasks. **Second**, LLM-IQA shows excellent performance across

various distortion tasks, *e.g.* natural scene blur, synthetic noise, and AI-generated semantic mistakes. Notably, our model has not undergone any task-specific training for these distortions. This indicates that our model has fully leveraged the zero-shot capabilities of MLLMs, enabling it to truly understand the assessment requirements of different distortion types and make accurate judgments, thus serving as a reliable assistant for a wide range of IQA tasks.

# E COMPLETE PROMPT

Here we provide the prompt used for number-standard, word-standard, and sentence-standard. As for word-standard, we provide the prompts with various numbers of words.

Below is the prompt for the 9-word standard.

> *# System: `` You are a helpful assistant to help me evaluate the quality of the image. You will be given standards about each quality level. The quality standard is listed as follows: 9: Perfect, 8: Excellent, 7: Very Good, 6: Good, 5: Fair, 4: Bad, 3: Poor, 2: Awful, 1: Very Bad. The higher the image quality, the higher the score should be. Please strictly follow the USER's format, otherwise the result will be invalid.*
> *# User: `` please evaluate the quality of the image and score in [1,2,3,4,5,6,7,8,9]. Only tell me the number. Do not analyze the image.*

Below is the prompt for the 7-word standard.

> *# System: `` You are a helpful assistant to help me evaluate the quality of the image. You will be given standards about each quality level. The quality standard is listed as follows: 7: Perfect, 6: Excellent, 5: Good, 4: Fair, 3: Bad, 2: Poor, 1: Very Bad. The higher the image quality, the higher the score should be. Please strictly follow the USER's format, otherwise the result will be invalid.*
> *# User: `` please evaluate the quality of the image and score in [1,2,3,4,5,6,7]. Only tell me the number. Do not analyze the image.*

Below is the prompt for the 5-word standard.

> *# System: `` You are a helpful assistant to help me evaluate the quality of the image. You will be given standards about each quality level. The quality standard is listed as follows: 5: Excellent, 4: Good, 3: Fair, 2: Bad, 1: Poor. The higher the image quality, the higher the score should be. Please strictly follow the USER's format, otherwise the result will be invalid.*
> *# User: `` please evaluate the quality of the image and score in [1,2,3,4,5]. Only tell me the number. Do not analyze the image.*

Below is the prompt for the 3-word standard.

> *# System: `` You are a helpful assistant to help me evaluate the quality of the image. You will be given standards about each quality level. The quality standard is listed as follows: 3: Excellent, 2: Fair, 1: Poor. The higher the image quality, the higher the score should be. Please strictly follow the USER's format, otherwise the result will be invalid.*
> *# User: `` please evaluate the quality of the image and score in [1,2,3]. Only tell me the number. Do not analyze the image.*

Below is the prompt for the 7-sentence standard.

> *# System:* `` *You are a helpful assistant to help me evaluate the quality of the image. You will be given standards about each quality level. The quality standard is listed as follows:*
> *7: Perfect! The overall quality of the image is unparalleled, with every detail meeting the highest standards.*
> *6: Excellent! The overall quality of the image is excellent, with all aspects being exemplary and without flaw.*
> *5: Good! The overall quality of the image is good, and it is satisfactory in many aspects.*
> *4: Fair! The overall quality of the image is fair. There are certain merits but also some deficiencies.*
> *3: Bad! The overall quality of the image is bad, with noticeable shortcomings.*
> *2: Poor! The overall quality of the image is poor, with obvious defects.*
> *1: Very Bad! The overall quality of the image is very bad and hard to accept.*
> *The higher the image quality, the higher the score should be.*
> *Please strictly follow the USER's format, otherwise the result will be invalid.*
> *# User:* `` *please evaluate the quality of the image and score in [1,2,3,4,5,6,7]. Only tell me the number. Do not analyze the image.*

Below is the prompt for the 7-number standard.

> *# System:* `` *You are a helpful assistant to help me evaluate the quality of the image. The higher the image quality, the higher the score should be. Please strictly follow the USER's format, otherwise the result will be invalid.*
> *# User:* `` *please evaluate the quality of the image and score in [1,2,3,4,5,6,7]. Only tell me the number. Do not analyze the image.*