# OpenReview forum: "LLM-IQA: Standard-guided MLLM for Mix-grained Image Quality Assessment"
_ICLR.cc/2026/Conference — Submitted to ICLR 2026_

### Official Review · Reviewer_kceR · 2025-10-26

**Soundness:** 2
**Presentation:** 2
**Contribution:** 3
**Rating:** 4
**Confidence:** 4

**Summary:**

The paper proposes LLM-IQA, a zero-shot, training-free image quality assessment method designed to address the poor out-of-distribution generalization and high training costs of existing IQA models (p. 1, abstract). The method mimics the human evaluation process (Fig. 2) and is implemented through two core mechanisms: (1) a standard-guided scoring mechanism (Sec. 3.1), which provides the MLLM with explicit quality level standards (e.g., “7: Perfect, 6: Excellent...”) and constrains the output to a single discrete token (an integer between 1 and 7); and (2) a mixed-granularity aggregation mechanism (Sec. 3.2), which scores both the global image and segmented object subregions, aggregating the final score via area-weighted averaging. Experimental results show that LLM-IQA outperforms all training-free methods across five datasets (Table 2) and achieves performance comparable to trained methods in cross-dataset scenarios (Table 5, pp. 6-7).

**Strengths:**

The main contribution lies in proposing a novel training-free IQA paradigm that effectively leverages the MLLM’s prior knowledge and alignment with human perception (p. 1). The standard-guided scoring mechanism (Sec. 3.1) is a clever design that transforms continuous score regression into a discrete, single-token classification task bound to descriptive vocabulary, which accords with the generative characteristics of MLLMs (Sec. 3.1 discusses the difficulty MLLMs have in producing precise floating-point numbers such as 87.5). Ablation studies (Table 3, Exp. 1 vs. 6) demonstrate the substantial performance improvement of single-token outputs over multi-token floating-point outputs (SRCC on SPAQ increases from 0.616 to 0.891). The mixed-granularity aggregation strategy, inspired by the human evaluation process (Fig. 2), combines global and local information; experiments (Table 3, Exp. 6, 7, 8) show that mixed granularity (SRCC 0.886) is superior to using either global (0.891) or local (0.851) information alone, which is a sound and effective design.

**Weaknesses:**

There is a critical internal contradiction in the methodological description that seriously affects reproducibility. Sec. 3.2, Fig. 3, and Algorithm 1 provide detailed descriptions of using “segmentation” and “masks” to extract local regions, yet the ablation study in Sec. 4.4 (Table 3, Exp. 5) explicitly reports a “significant drop” in performance when using the “Mask” setting (SRCC on SPAQ is only 0.632), attributed to the MLLM’s visual encoder misinterpreting the masked fill regions. In contrast, the superior settings (Exps. 7, 8, 9) all employ “BBox” (bounding boxes), and the paper even asserts on p. 8 that “applying bounding boxes as the segmentation method is necessary.” This fundamental conflict between the methodological (Sec. 3) and experimental (Sec. 4.4) discussions regarding the use of “masks” versus “bounding boxes” leaves readers uncertain about the actual implementation of the final model. In addition, Table 5 shows relatively lower performance on the synthetic dataset KADID-10k (SRCC 0.671), which the paper attributes to distributional differences (p. 7), potentially revealing limitations in the MLLM’s prior knowledge that predominantly stems from natural images.

**Questions:**

Please clarify definitively whether the local scoring in the "Mix-Grained Aggregation" mechanism is based on "Masks" or "BBox" (Bounding Boxes). The ablation study in Table 3 shows that using "Mask" (Exp 5) results in poor performance, while "BBox" (Exp 7) performs much better. This contradicts the detailed descriptions in Section 3.2, Figure 3, and Algorithm 1, which all focus on "segmentation" and "masks."

In Section 4.4, the paper mentions an "attention scheme" to explain the effectiveness of the "area-weighted average". However, based on the paper's description, the method appears to be a simple weighted sum rather than incorporating an explicit attention module. What exactly does "attention" refer to in this context?

The paper notes in Section 4.2 and Table 5 that the model's performance on the KADID-10k dataset is relatively low, attributing this to its "synthetic" distribution difference. Does this expose a fundamental limitation of LLM-IQA, suggesting that the MLLM's prior knowledge is primarily derived from natural images, leading to a limited ability to assess specific types of synthetic distortions?

Table 5 (Segmentation Setting) demonstrates the necessity of the "complement mask". This part typically corresponds to the image background. How is its quality score specifically combined with the object (foreground) scores in the "Mix-Grained Aggregation"? Given that the background area can be very large, does it disproportionately dominate the final local quality score in the area-weighted average?

In the "Standard-Guided Scoring" mechanism, why was K=7 chosen as the optimal number of levels (Table 4)14141414? The paper notes that performance drops when K=9. Does this imply that when the descriptive standards (words) become too dense, the MLLM's ability to distinguish between them decreases, thus causing this performance turning point?

---

### Official Review · Reviewer_Jtdm · 2025-10-27

**Soundness:** 2
**Presentation:** 1
**Contribution:** 2
**Rating:** 2
**Confidence:** 5

**Summary:**

This paper introduces LLM-IQA, a training-free pipeline for image quality assessment. It achieves state-of-the-art performance among training-free approaches. The method segments an input image into sub-images centered on salient objects, evaluates them with an MLLM, and aggregates the results into a single quality score.

**Strengths:**

1. The proposed training-free method achieves state-of-the-art performance compared with other approaches.
2. The method is straightforward to understand and implement.
3. The paper adopts an approach that identifies the main objects and then performs regression to obtain the final score. This method is simple, straightforward, and intuitive.

**Weaknesses:**

1. The paper lacks innovation. Large models can achieve much more than just scoring, so if the focus is solely on scoring, what is the motivation for adopting a training-free approach? Is it meant to achieve better generalization, or does a training-free method offer greater robustness in new scenarios? From Figure 5, the results do not appear particularly impressive.
2. The accuracy of the local quality predictions is uncertain, as supporting results are not provided.
3. The paper has many writing issues. For example, Table 5 is mistakenly referred to as Figure 5, and DogIQA appears in the figures without any prior mention in the text. The citation formatting in the references also has major problems.
4. The cross-validation experiments are insufficient, as all training is conducted on real distortion scenarios. The generalization experiments are limited to only a few cases and should include more diverse evaluations, such as testing on synthetic distortion datasets like SRIQA-Bench, to better assess the method’s robustness and applicability.
5. The work provides few insights and appears to have limited significance for the image quality assessment field.

**Questions:**

See Weakness.

---

### Official Review · Reviewer_ziYU · 2025-10-29

**Soundness:** 2
**Presentation:** 2
**Contribution:** 3
**Rating:** 2
**Confidence:** 4

**Summary:**

This paper introduces a multimodal large language model (MLLM)-based image quality assessment (IQA) inference pipeline that imitates human evaluators' scoreing process, because IQA still suffers from poor out-of-distribution generalization ability.
First, a standard quality level text guided scoring prompt to boost IQA score estimation is applied to the proposed LLM-IQA.
Second, a segmentation model is adopted to evaluate object-centered sub-images as well as the whole image.
The final score is computed by aggregating the local and global scores.
In experiments, LLM-IQA shows comparable results without fine-tuning or task-specific training for the IQA task.

**Strengths:**

The authors propose three components to effectively predict IQA scores using MLLM, and validate the effectiveness of each through an ablation study (Table 3).
- Represent image quality using a single token to achieve an accurate score (Insight 1).
- A combination of text and numbers is a more effective prompt format for MLLM-based IQA (Insight 2).
- Local and global score aggregation.

**Weaknesses:**

Major flaw: There are inconsistencies in the methodological descriptions.
- L263 states that a model ensemble was used, but the implementation details (L290) indicate that a single mPLUG-Owl3 model was used. The paper should clarify what is meant by "model ensemble" in this context.
- In the ablation study (Table 3), Experiment index 9 reports results for ensemble, but the methodology and experimental procedure about ensemble are not described.
- Moreover, if the results in Section 4.2 also derived from the ensemble setting, readers are not informed of this, which undermines the overall reliability of the experimental results.

Missing recent training-based IQA in related work and experimental results
- (NIPS 2024) Adaptive image quality assessment via teaching large multimodal model to compare
- (CVPR 2024) Q-Instruct: Improving Low-level Visual Abilities for Multi-modality Foundation Models
- (CVPR 2025) Teaching large language models to regress accurate image quality scores using score distribution
- (arxiv 2025) Q-insight: Understanding image quality via visual reinforcement learning

Typos
- Figure 4: What is $s_{Dog}$?
- L369
- Figure 5 should be Table 5.

**Questions:**

I wonder whether it would be possible to compare with the latest training-free IQA published after CLIP-IQA.

---

### Official Review · Reviewer_mq7X · 2025-10-31

**Soundness:** 2
**Presentation:** 2
**Contribution:** 2
**Rating:** 2
**Confidence:** 5

**Summary:**

This paper proposes LLM-IQA, a zero-shot, training-free image quality assessment method. The pipeline involves segmenting an image, using a multimodal large language model to score the global image and local patches against a predefined standard, and aggregating scores using an area-weighted average. The authors present experiments showing that LLM-IQA achieves state-of-the-art performance among training-free methods and competitive performance with training-based models in cross-dataset evaluations.

**Strengths:**

1. The work correctly identifies a critical challenge in the IQA domain: the out-of-domain generalization of training-based models due to the prohibitive cost and labor involved in creating large-scale labeled datasets. A robust zero-shot, training-free method would be a valuable contribution.

2. The proposed method demonstrates SOTA performance when compared to other training-free IQA models (like BRISQUE and NIQE) across five diverse datasets. Furthermore, it shows competitive performance against supervised models in cross-dataset evaluations.

**Weaknesses:**

1. The primary weakness is the paper's limited novelty. The proposed method, which combines segmentation, discrete-level prompting, and score aggregation, appears to be more an application of sophisticated prompt engineering than a novel IQA framework.
1) The strategy of prompting an MLLM with predefined standards and averaging scores from multiple inferences on local patches is not new. For instance, Wen et al. [1] employ a very similar strategy for video quality assessment. Other zero-shot MLLM benchmarks, such as Q-Bench [2], also propose methods for converting token logits to scores, which serve a similar purpose. The paper fails to differentiate its approach from these existing techniques.
2) The contribution seems tied to the specific MLLM used (mPLUG-Owl3). To validate the pipeline as the main contribution, its efficacy should be demonstrated across different MLLMs. As is, it is difficult to separate the performance of the LLM-IQA framework from the strong internal perceptual capabilities of the base model.

2. The captions for the ablation studies are not self-contained. For example, Table 3 lists settings like "Aggregation: Area" or "Standard: Word" without explanation. A reader must hunt through Section 4.4  to understand what is being compared.

3. There is a significant error in the labeling of "Figure 5". This "figure" is, in fact, the main results table for the training-based model comparison. Compounding this, the text in Section 4.2 repeatedly refers to this data as "Table 5", which contradicts the actual Table 5 on page 9. This confusion is substantial.

[1]  Wen, Wen, et al. "An Ensemble Approach to Short-form Video Quality Assessment Using Multimodal LLM." ICASSP, 2025.

[2] Wu, Haoning, et al. "Q-Bench: A Benchmark for General-Purpose Foundation Models on Low-level Vision." ICLR, 2024.

**Questions:**

1. The datasets used for the ablation studies are inconsistent. Table 3  uses SPAQ, AGIQA-3k, and LIVE Challenge. Table 4  uses SPAQ, KADID-10k, and AGIQA-3k. Table 5 reverts to the set from Table 3. For a rigorous analysis, all ablation experiments should be reported consistently across all major evaluation datasets.
2. The comparison to training-based methods is incomplete. Several recent and highly relevant MLLM-based IQA models are known for their strong generalization, such as Q-Insight [3] and VisualQuality-R1[4]. These represent the current state-of-the-art and should be included as competing models.

3. More IQA datasets, such as PIPAL, can also be added to further validate the claims
of OOD robustness.

[3] Li, Weiqi, et al. "Q-insight: Understanding image quality via visual reinforcement learning." Neurips, 2025.[4] Wu, Tianhe, et al. "VisualQuality-R1: Reasoning-Induced Image Quality Assessment via Reinforcement Learning to Rank." Neurips, 2025.

---

### Meta-Review · Area_Chair_fpH9 · 2026-01-06

**Summary:**

This paper proposes a zero-shot, training-free image quality assessment method, i.e., LLM-IQA, with three main components: segmenting an image, using a multimodal large language model to score the global image and local patches against a predefined standard, and aggregating scores using an area-weighted average. The reviewers pointed out several critical concerns regarding different aspects of this paper, including but not limited to:

1. The technical novelty and contribution are limited. The proposed method combines segmentation, discrete-level prompting, and score aggregation, but it appears to be an application of sophisticated prompt engineering instead of a novel IQA framework.
2. The proposed method is tied to the specific MLLM used, i.e., mPLUG-Owl3. The efficacy should be demonstrated across different MLLMs.
3. The comparison with existing methods is far from enough. Some previous methods have used the strategy of prompting an MLLM with predefined standards and averaging scores from multiple inferences on local patches. The paper fails to differentiate its approach from the existing techniques. Some recent training-based IQA methods are missing in related work and experimental results.
4. There are inconsistencies in the methodological descriptions. For example, the captions for the ablation studies are not self-contained.
5. The accuracy of the local quality predictions is uncertain, as supporting results are not provided. The cross-validation experiments are insufficient, as all training is conducted on real distortion scenarios.
6. The work provides few insights and appears to have limited significance for the IQA field.

**Reviewer Concerns:**

No rebuttal was provided, so the answer for this question is: all review concerns are still outstanding.

**Reviewer Scores:**

None.

---

### Decision · Program_Chairs · 2026-01-26

Reject